# Work-Related Exposures and Sickness Absence Trajectories: A Nationally Representative Follow-up Study among Finnish Working-Aged People

**DOI:** 10.3390/ijerph16122099

**Published:** 2019-06-13

**Authors:** Tea Lallukka, Leena Kaila-Kangas, Minna Mänty, Seppo Koskinen, Eija Haukka, Johanna Kausto, Päivi Leino-Arjas, Risto Kaikkonen, Jaana I. Halonen, Rahman Shiri

**Affiliations:** 1Department of Public Health, University of Helsinki, P.O. Box 20, 00014 Helsinki, Finland; minna.manty@helsinki.fi (M.M.); risto.kaikkonen@helsinki.fi (R.K.); 2Finnish Institute of Occupational Health, P.O. Box 40, 00032 Helsinki, Finland; leena.kaila-kangas@ttl.fi (L.K.-K.); eija.haukka@ttl.fi (E.H.); johanna.kausto@ttl.fi (J.K.); paivi.leino-arjas@ttl.fi (P.L.-A.); jaana.halonen@ttl.fi (J.I.H.); rahman.shiri@ttl.fi (R.S.); 3City of Vantaa, Asematie 7, 01300 Vantaa, Finland; 4Unit of Statistics and research, National Institute for Health and Welfare, Helsinki, P.O. Box 30, 00271 Helsinki, Finland; seppo.koskinen@thl.fi

**Keywords:** occupational cohort, register-based, work disability, sedentary, physical heaviness, prospective

## Abstract

The contribution of physically demanding work to the developmental trajectories of sickness absence (SA) has seldom been examined. We analyzed the associations of 12 physical work exposures, individually and in combination, with SA trajectories among the occupationally active in the Finnish nationally representative Health 2000 survey. We included 3814 participants aged 30–59 years at baseline, when exposure history to work-related factors was reported. The survey and interview responses were linked with the annual number of medically confirmed SA spells through 2002–2008 from national registries. Trajectory analyses identified three SA subgroups: 1 = low (54.6%), 2 = slowly increasing (33.7%), and 3 = high (11.7%). After adjustments, sitting or use of keyboard >1 year was inversely associated with the high SA trajectory (odds ratio, OR, 0.57; 95% 95% confidence interval, CI, 0.43–0.77). The odds of belonging to the trajectory of high SA increased with an increasing number of risk factors, and was highest for those with ≥4 physical workload factors (OR 2.71; 95% CI 1.99–3.69). In conclusion, these findings highlight the need to find ways to better maintain the work ability of those in physically loading work, particularly when there occurs exposure to several workload factors.

## 1. Introduction

Physical work-related factors have been linked to the risk of both sickness absence (SA) [1,2,3,4,5,6] and disability retirement [1,2,7,8]. However, the evidence is still inconclusive and could depend on the used exposures, age groups, and methods. While most studies have focused on adverse effects of physical work, some factors could also be protective of work ability, or decrease the risk of SA, but this has rarely been considered. Although excessive sitting or sedentariness are often linked to adverse health outcomes such as cardiovascular diseases [9], people who have sedentary or light work, versus those with physically heavy work, do not have a higher risk of musculoskeletal health outcomes [10,11,12], although the evidence is somewhat inconsistent [13]. There is less evidence regarding SA, but an intervention program found no differences between a decrease in sitting time and SA. [14]. Additionally, it is not clear if sitting increases or lowers the risk of SA, when it occurs with otherwise physically heavy work. Indeed, another important gap in evidence is that the different work-related factors have rarely been studied in combinations in relation to the risk of SA. Scarce evidence exists that exposure to more than one risk factor is likely to increase the risk of SA [15,16].

Previous studies have often either measured current work exposures, or with a gap between repeated measurements, without possibilities to confirm the significance of long-term exposure to SA [4]. Both short and long term exposure to physically heavy work during work history increased the risk of long SA in a Danish study [1]. While the study covered work histories for more than 20 years, SA data were available only for SA periods lasting above 30 days, and the data only comprised older workers. Overall, previous studies have not typically distinguished between work-related exposures during earlier and later careers, and have mostly included midlife and older employees. Although SA is common already among young employees, there is little evidence available on the contribution of physical workload on the development of SA using representative data of working populations with a wider age range.

Person-oriented methods have only rarely been applied when examining associations between physical work-related factors and SA. One study that applied a trajectory analysis for SA focused on kitchen workers examined self-reported SA due to musculoskeletal disorders [17]. In another previous study in the same data set as the present, the main focus was on pain, and physical workload was but a dichotomized covariate [18]. Previous evidence thus is largely from studies about the associations among variables (work exposures and e.g., dichotomous outcomes or count data), whereas in a person-oriented approach, the focus is on identifying latent groups of individuals who share similar developmental pattern in their SA over time [19,20]. After the developmental trajectories have been identified, work-related factors are used as predictors of trajectory group membership.

To fill in the gaps in evidence on the more detailed work-related exposures in relation to the long-term developmental trajectories of SA, we first identified SA trajectories over a 7-year follow-up using nationally representative data. Second, we examined whether 12 work-related factors, individually and cumulatively, increase or decrease the risk of SA, in terms of group memberships in the identified SA trajectories.

## 2. Materials and Methods

### 2.1. Participants

Data for this study were survey and register based. First, we used baseline data from the nationally representative Health 2000 Survey, where participants represented the demographic distributions of the Finnish adult population [21,22]. The Statistics Finland planned a 2-stage stratified cluster sampling design. The interviews were started on the 15 August 2000, and the health examinations on the 18 September 2000, and they continued until mid-June 2001, yielding a total participation rate of 89% [22]. The in-home interviews and several questionnaires comprised data on physical working conditions as well as several social and health related covariates. For this study, we included participants who were 30–59 years old at baseline. Our focus was on SA trajectories, and these can only be studied among those of working age and economically active. Moreover, health examinations by field physicians were only made for the participants who were 30 years or older, among whom we then were able to control the analyses for health variables. Thus, for the final sample, we included 1791 men and 2023 women. Further details of the data collection are available elsewhere [22], and at https://thl.fi/en/web/thlfi-en/research-and-expertwork/projects-and-programmes/health-2000-2011.

### 2.2. Ethical Approvals

The study protocol of the Health 2000 Survey has been ethically approved by the Ethics Committee for Epidemiology and Public Health of the hospital district of Helsinki and Uusimaa in Finland. Questionnaire survey data were prospectively linked to register based SA data. All the participants signed their written informed consent also for future registry linkages.

### 2.3. Physical Work-Related Exposures

Information on physical work-related exposures was collected in home interviews in 2001. Participants reported whether they were exposed daily to 12 exposures (no/yes) in their current or past five jobs. The first two of the 12 exposures were considered as factors that could potentially decrease the risk of SA: prolonged sitting excluding occupational driving (for 5 h or more), prolonged keyboard use (for 4 h or more), prolonged standing or walking (for 5 h or more), work requiring high handgrip force (3 kg per hand for 1 h or more), repetitive arm movement (for 2 h or more), using a vibrating tool (for 2 h or more), frequent manual handling of loads (more than 5 kg for 2 h or more at least 2 times per minute), manual handling of loads (more than 20 kg at least 10 times during a work day), squatting or kneeling (for 1 h or more), working in bent postures (for 1 h or more), working with the arms above shoulder level (for 1 h or more), and strenuous physical work in general, that included lifting or carrying heavy objects, excavating, digging and pushing. The duration of exposure to each of these work factors were reported (in years) and classified into the following groups: (1) No exposure, (2) 1−15 years, and (3) more than 15 years.

Based on the preliminary results, we formed additional summary variables: (1) factors that could decrease the risk of SA (sitting and computer work combined) and (2) factors that could increase the risk of SA, i.e., exposure to the nine work-related risk factors, classified into four groups: 0, 1, 2–3 or 4 or more work-related exposures. The variable of overall strenuousness of work was omitted from the summary measure. For the analysis examining combination of factors that could increase or decrease the risk of SA, we further formed a variable measuring combination of exposures: neither of the summary variables; only factors that could decrease the risk of SA; only factors that could increase the risk of SA; or both potentially risk decreasing and increasing factors.

### 2.4. Register Data

SA data were obtained from the Social Insurance Institution of Finland that registers absence periods over nine days from all employers [23]. All SA periods in each year during the follow-up from 2002 to 2008 were included in the trajectory analyses except if the participant had retired or died, when the follow-up ended in the beginning of the year of the event. Thus, all participants contributed to the trajectories for each complete follow-up year, provided they were part of the workforce for the entire year. The number of spells per year varied between 0 and 4 in each follow-up year. Data on retirement events were provided by the Finnish Centre for Pensions and data on deaths were obtained from Statistics Finland. Register data were linked to the Health 2000 data by each participant’s personal identification number. We received the anonymized data without any identity codes.

### 2.5. Covariates

From the interviews, questionnaires and physical examinations, we included information on age, gender, socioeconomic and health-related factors. Based on the years of basic education, the participants were divided into three groups: low (≤9 years), intermediate (10–12 years) and high (≥13 years). Marital status was dichotomized into single vs. married/cohabiting. Weight and height were measured and body mass index (BMI, kg/m^2^) was classified into three groups: normal weight, overweight, and obese. Current daily smoking (no/yes) was enquired in the interview. Alcohol use disorders (dependence and abuse) were diagnosed using the Composite International Diagnostic Interview(CIDI) interview [22,24,25]. There were two categories for leisure time physical activity: exercising at least once a week (active)/more seldom (passive). Sleep problems were inquired by one question and responses were dichotomized (no/yes).

Psychosocial strain was measured using the Job Content Questionnaire [26]. The scales of work demands comprised five items (Cronbach’s alpha, α = 0.79), and job control nine items (α = 0.84). Both variables were dichotomized at their median, combined and classified into two categories (high job strain/no strain). Participants were asked whether they received support from their supervisors (two questions) and from their co-workers (two questions), when needed. The response alternatives were 1 = fully agree, 2 = quite agree, 3 = do not agree or disagree, 4 = quite disagree, and 5 = completely disagree. The scales were classified and merged into high (1−2) and low (3−5) social support.

Musculoskeletal disorders (M00–99) were diagnosed in the clinical examination by a physician, based on disease history, symptoms, and clinical findings (17). The participants were categorized as having a chronic disease, if a physician diagnosed one of the following: cardiovascular-, respiratory- or neurological disease, diabetes, cancer, peptic ulcer or permanent injury. The participants were categorized as having a mental disorder (no/yes), if a physician diagnosed one of the following: psychosis, depression or anxiety.

### 2.6. Statistical Analyses

Trajectory analysis was used to identify latent groups (trajectories) of participants having a similar developmental pattern in their SA over time. This semiparametric approach uses maximum likelihood methods to estimate probabilities for trajectories and fits well to longitudinal data. The annual number of SA periods was modelled using zero inflated Poisson distribution (link function Zero Inflated Poisson, ZIP). This link function was chosen because our outcome was based on the number of SA periods per year, which does not follow the Gaussian distribution. If the person had several SA periods, they affected the probability of the trajectory membership. The participants were assigned to the trajectory to which they had the highest probability to belong to. Selection of the optimal number of trajectories and their shapes were based on the Bayesian information criterion (BIC). Model selection and fit statistics are displayed in Appendix A. SA trajectories were analyzed by Proc Traj. [27,28]. More details of the method are available elsewhere [29]. As trajectories were similar regarding their shape for both younger and older employee groups (Appendix A), the main trajectory and subsequent analyses were conducted in pooled data, and associations between work-related factors and trajectory group memberships are only displayed for all participants, adjusting for age. Moreover, the associations between work-related factors and trajectory memberships would also largely have been under-powered in the age stratified analysis. Associations between history of exposure to the 12 work-related factors and the trajectory membership were assessed using multinomial logistic regression, with the low trajectory as the reference category. These models were adjusted for age (continuous), gender, basic education, marital status, BMI, smoking, leisure time physical activity, alcohol dependence, job strain, social support at work, sleep problems, musculoskeletal disorders, other physical diseases, and mental disorders. We used the SAS software package (version 9.4; SAS Institute, Inc., Cary, NC, USA) for all statistical analyses. 

## 3. Results

We identified three distinct SA trajectories over the follow-up: 1 = low (54.6% of participants), 2 = slowly increasing (33.7%), and 3 = high (11.7%) (Figure 1). At baseline, the mean age was 43.4 years (standard deviation, SD 8.0) among all participants, 42.6 years (SD 8.1) in the low trajectory group, 44.1 years (SD 7.8) in the slowly increasing trajectory group, and 45.4 years (SD 7.1) in the high trajectory group.

There were clear differences in the distributions of sociodemographic and health-related determinants between the three identified trajectory groups (Table 1). The proportion of men was 50% in the low SA trajectory, and 35% in the high. Low education as well as practically all behavioral risk factors, such as obesity and smoking, and health-related factors, such as musculoskeletal disorders, were also linked to the membership of the high SA trajectory.

Next, we examined how the history of exposure to the various work- related factors that could decrease the risk of SA, as well as physical work exposures that could increase the risk, associated with trajectory memberships (Table 2). Prolonged sitting and keyboard use were inversely associated with memberships in both the slowly increasing and high SA trajectories, but only among those who had been exposed up to 15 years. There were some differences for shorter (1–15 years) and longer (more than 15 years) exposures, in the full models that simultaneously considered all social and health-related determinants of SA.

Regarding the risk factors, all examined exposures were associated with the high SA trajectory, and some also with the slowly increasing SA trajectory. There was variation in the contribution of the duration of the exposure, i.e., sometimes the associations were statistically confirmed for shorter exposure (1–15 times/years) for the slowly increasing trajectory only. In contrast, long exposure (>15 years) to physical factors was associated with the membership of the high trajectory. An exception was working with arms above shoulder level, where the association was observed only for the shorter exposure time.

Table 3 displays the associations for the summary variables. For those reporting either prolonged sitting or keyboard use, or both the odds of belonging to the slowly increasing or the high SA trajectory was lower compared to those reporting neither of these factors. Further, it was observed that the higher the number of risk factors reported, the higher were the odds of belonging to the slowly increasing or high SA trajectory groups. The odds increased even with one risk factor and it was the highest for those reporting four or more risk factors (OR 2.71; 95% CI 1.99–3.69).

Finally, reporting only sitting or prolonged keyboard use was associated with lower odds of belonging to the trajectory of high SA, while reporting only physically demanding factors was associated with higher odds of belonging to the trajectory of high SA (Table 4). Combination of both types of factors did not increase the odds of belonging to the trajectory of high SA.

## 4. Discussion

### 4.1. Main Findings

This study identified three distinctive SA trajectories: low, slowly increasing and high, among a follow-up of working-aged Finns in a nationally representative sample. Long-term exposure to high physical workload factors increased the risk of membership in the group of high SA trajectory, and the risk was the higher the higher the number of exposures. On the contrary, prolonged sitting and keyboard use were associated with a lower likelihood of belonging to the high SA trajectory. However, exposure to sitting or keyboard use for more than 15-years was not associated with lower odds of membership in the high SA trajectory. Finally, reporting work-related physical risk factors in combination with sitting or keyboard use was not associated with the membership of the high SA trajectory.

### 4.2. Interpretation

Our finding about the importance of cumulative exposure to several physical workload factors is in line with a previous study, which reported that a higher number of different workload factors was associated with an increased risk of SA in Denmark [16]. However, in that study the associations between work exposures and incident SA during the follow-up were assessed using Cox regression and thus it could not identify development of SA over time or latent groups in the data. Neither were sitting or other potentially protective factors included. Thus, it was not possible to confirm, if some work-related factors decreased the incidence of SA, or whether the increased incidence concerned those with physical exposures only. Nonetheless, these nationally representative Nordic studies highlight the importance of focusing on cumulative contributions of different physical workload factors, as employees with multiple exposures are at a particularly high risk of SA.

Findings concerning sitting and keyboard use, and the associated decreased risk of SA, could be seen as both contrasting with and adding to the previous evidence regarding other outcomes. While sedentary behaviors have been linked to adverse health outcomes, these mainly refer to cardiovascular diseases [9], which are not a common reason for SA. Moreover, the results should not be directly compared, as we addressed occupational sitting as a risk/predictive factor, but not other sedentary behaviors such as watching television which could explain the differences in the findings between ours and previous studies on sedentary behaviors and other health outcomes. Our results further contrast those of a Swedish cross-sectional study, which reported low exposure to seated work to be associated with lower odds of excellent work ability among older workers with neck pain [30]. However, the outcomes are not directly comparable, as excellent work ability was self-reported, while we focused on register-based SA trajectories over a 7-year period after the assessment of the exposure. People with neck pain may react differently to the examined exposures as compared to employees without such pain. Although one may assume that sitting and keyboard use mainly concern white-collar employees, who in general have a decreased risk of work disability as compared to manual workers with more physically demanding work [31,32], the protective effects remained after adjusting for socioeconomic and health-related factors. These potentially protective effects should be further explored and corroborated in other studies. As our study was observational and relied on self-reported exposure data on prolonged sitting, a protective factor would be a too strong term to be used. Rather, the interpretation is that that the inverse association could also be due to other unmeasured factors, and for example a randomized controlled trial might show a different result.

Finally, our additional analyses stratified by age group suggested that the associations were slightly stronger among younger versus older employees (data not shown). However, statistical power was low, and the results should be interpreted as indicative. Some differences in the associations between younger and older employees could be expected [33], but including older age groups could also induce bias due to selection. Indeed, it is possible that the most robust older employees had continued in their heavy work, while others had succumbed to illness or exited paid employment e.g., after a long-term SA to disability pension. This might have happened even before the collection of the baseline data. Such selection is supported by an earlier study using the same data, where participants who had a history of physically heavy work and had exited paid employment, had a higher risk of sciatica [12]. Healthy worker effect could have made our results more conservative.

### 4.3. Methodological Considerations

This study has some limitations and strengths that need to be acknowledged and discussed. First, the data regarding physical work-related factors were self-reported, and questions about the duration of exposure are likely to induce some memory bias. Common methods bias is, however, unlikely, as our outcome data, i.e., SA periods, were based on national registries of high reliability and the follow-up began only after the assessment of exposure. Additionally, group level data using a job exposure matrix of physical exposures, based on occupational titles, have produced results similar to the self-reported exposures for work disability outcomes [7].

Second, our outcome comprised all-cause SA periods. Physically demanding work could increase particularly the risk of work disability related to, say, musculoskeletal disorders [7,8]. However, diagnostic groups have often been combined also in previous studies, and the focus has been on all-cause SA [4,5]. While it is possible that the associations reflect those for musculoskeletal outcomes, particularly, physical work can also increase the risk of mental disorders [34,35]. This justifies a focus on all-cause SA trajectories. One may also question, if musculoskeletal disorders should be adjusted for, if sickness absence is largely due to these diagnoses, i.e., should such an adjustment be considered as over-adjustment. We have, however, conducted additional analyses without adjustment for musculoskeletal disorders, and the results were very similar (please see Appendix A which repeat analyses of the main Table 3 and Table 4 without adjustment for musculoskeletal disorders). Overall, the odds ratios for sitting and computer work attenuated slightly, and those for physical work exposures strengthened. However, statistical significance did not change for any of the exposures except for bent postures (all odds ratios were significant). Thus, over-adjustment should not be a major issue, or bias the examined associations. As we had several predictors and present different sets of analyses, we retained full models in the main tables, considering key pertinent risk factors of sickness absence.

Third, while the trajectory analysis is a useful tool to identify homogenous subgroups following similar developmental patterns of the outcome (in our case SA), it is important to note that any individual might follow a different trajectory. In other words, misclassifications are possible, and trajectories are approximations of the true development. The proportions of those with a posterior probability below 0.7 of belonging to a trajectory was 13.9%, which means that misclassification cannot be ruled out. However, the mean posterior probabilities were high. In addition, trajectory modelling was done following people until their retirement or exit from the cohort for other reasons such as death or emigration. This means that the number of follow-up time points varied, and the shape of the trajectory of the excluded ones cannot be confirmed. However, we conducted sensitivity analyses where we retained the same number of follow-up points for all, i.e., those who left the cohort were excluded. This resulted in lower numbers and some selection of participants, but the findings remained broadly similar. A key strength of this study is the inclusion of a nationally representative cohort, where it was possible to focus on the associations among people from all employment sectors, men and women, and duration over the majority of the working life span. Thus, we could include individuals in their earlier and later careers and confirm the contribution of the exposures to the development of SA among all employees. Another strength was the opportunity to include several different exposures and assess their cumulative effects on SA trajectories. Furthermore, we could examine and identify both risk factors of SA, factors that might decrease the risk of SA, and their combined contributions to SA. Finally, we could control the associations for social determinants of SA, health behaviors and medical conditions.

## 5. Conclusions

Physical work was associated with the high SA trajectory, with the highest risk found for those with cumulative exposure to heavy physical work. Sitting and keyboard use without physically heavy tasks were associated with a decreased risk of SA. Thus, the findings of this study provide no evidence that prolonged sitting at work would increase the risk of SA. Furthermore, the risk of belonging to the high SA trajectory concerned mainly those who only have physically heavy work, i.e., who do not also report sitting or keyboard use. As all the risks remained after controlling for various pertinent risk factors, these findings highlight the need to find ways to better maintain work ability of those with the physically most strenuous work.

## Figures and Tables

**Figure 1 ijerph-16-02099-f001:**
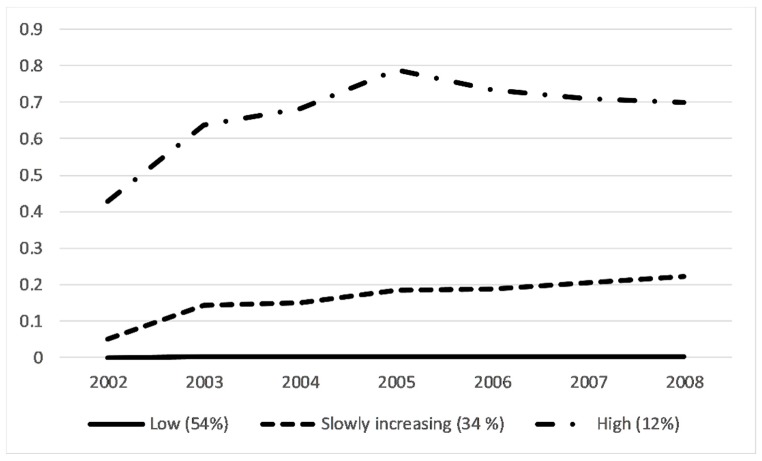
Sickness absence trajectories among 30–59-year old participants of the Health 2000 survey: 1 = low (54.6%), 2 = slowly increasing (33.7%), and 3 = high (11.7%) (x-axis: the follow-up from 2002 through 2008, y-axis = annual number of sickness absence periods).

**Table 1 ijerph-16-02099-t001:** Background characteristics of participants in three sickness absence trajectory groups.

Background Characteristics	All	Low		Slowly Increasing		High	
	*N* = 3814	*N* = 2083	%	*N* = 1287	%	*N* = 444	%
Gender, men	1791	1062	50.1	574	44.6	155	34.9
Marital status, single (vs. married/co-habiting)	857	472	22.7	277	21.5	108	24.3
Basic education							
high (> 13 years)	1831	1102	52.9	580	45.1	149	33.6
intermediate (10−12 years)	1233	613	29.4	449	34.8	171	38.5
low (<9 years)	750	368	17.7	258	20.1	124	27.9
Body mass index							
≤24.9 (normal)	1646	969	46.5	534	41.5	143	32.2
25−29.9, (overweight)	1479	812	39.0	489	38.0	178	40.1
≥30 (obese)	689	302	14.5	264	20.5	123	27.7
Daily smoking, yes	999	478	23.0	363	28.2	158	35.6
Alcohol dependence, yes	188	91	4.4	62	4.8	35	7.9
Leisure time physical activity, passive	936	495	23.8	303	23.5	138	31.1
Sleep problems, yes	2116	1068	51.3	741	57.6	307	69.1
Job strain, yes	552	263	12.6	196	15.2	93	21.0
Social support at work, low	1358	723	34.7	468	36.4	167	37.6
Musculoskeletal disorders, yes	1091	472	22.7	416	32.3	203	45.7
Mental disorders, yes	314	124	6.0	129	10.0	61	13.7
Any other diseases, yes	1361	662	31.8	481	37.4	218	49.1

**Table 2 ijerph-16-02099-t002:** Fully adjusted associations of physically demanding work factors with sickness absence trajectories (*N* = 3814). Odds ratios (OR) from multinomial regression analyses with the low sickness absence trajectory as reference.

Physically Demanding Work Factors	All	Low	Slowly Increasing	High	Full Model *		Full Model *	
					Trajectory Slowly Increasing vs. Low		Trajectory High vs. Low	
Reference = 1	*N*	*N*	*N*	*N*	OR ^α^	95% CI ^β^	OR ^α^	95% CI ^β^
Physical exposure in years								
Prolonged sitting								
0	2411	1240	858	313	1		1	
1−15	876	566	247	63	0.66	0.55−0.80	0.48	0.36−0.66
>15	527	277	182	68	0.86	0.69−1.07	0.86	0.62−1.18
Prolonged keyboard use								
0	2766	1469	951	346	1		1	
1−15	674	416	205	53	0.78	0.64−0.95	0.55	0.40−0.77
>15	374	198	131	45	0.91	0.71−1.17	0.81	0.56−1.18
Prolonged standing or walking								
0	2102	1249	664	189	1		1	
1−15	1009	507	363	139	1.42	1.19−1.68	1.87	1.44−2.42
>15	703	327	260	116	1.42	1.16−1.73	2.09	1.58−2.78
Repetitive arm movement								
0	2232	1267	750	215	1		1	
1−15	925	507	304	114	1.02	0.86−1.22	1.26	0.96−1.64
>15	657	309	233	115	1.18	0.96−1.44	1.87	1.41−2.48
Arms above shoulder level								
0	3095	1726	1035	334	1		1	
1−15	380	187	137	56	1.22	0.96−1.55	1.45	1.03−2.05
>15	339	170	115	54	0.99	0.76−1.28	1.26	0.88−1.81
Bent postures								
0	2764	1569	921	274	1		1	
1−15	571	288	194	89	1.15	0.94−1.41	1.70	1.29−2.27
>15	479	226	172	81	1.18	0.94−1.48	1.70	1.25−2.32
Squatting or kneeling								
0	2991	1696	975	320	1		1	
1−15	448	214	169	65	1.46	1.16−1.82	1.71	1.23−2.37
>15	375	173	143	59	1.39	1.09−1.78	1.72	1.22−2.44
Using a vibrating tool								
0	3553	1943	1209	401	1		1	
1−15	122	68	37	17	0.97	0.64−1.48	1.50	0.83-2.69
>15	139	72	41	26	0.93	0.62−1.40	2.06	1.23−3.45
Work, that requires high hand grip force								
0	3007	1696	994	317	1		1	
1−15	247	116	93	38	1.37	1.08−1.73	1.69	1.20−2.39
>15	560	271	200	89	1.27	1.00−1.63	1.82	1.30−2.54
Frequent handling of loads at least 5 kg								
0	3267	1826	1094	347	1		1	
1−15	305	143	113	49	1.35	1.03–1.76	1.86	1.28−2.69
>15	242	114	80	48	1.14	0.84−1.55	2.05	1.38−3.03
Handling of loads of at least 20 kg								
0	3202	1796	1054	352	1		1	
1−15	332	157	128	47	1.53	1.18−1.97	1.73	1.19−2.51
>15	280	130	105	45	1.36	1.03−1.80	1.65	1.11−2.44
Strenuous physical work overall								
0	2816	1609	925	282	1		1	
1−15	542	259	200	83	1.40	1.14−1.73	1.85	1.37−2.49
>15	456	215	162	79	1.26	1.01−1.59	1.82	1.32−2.51

^α^ Odds ratio, ^β^ 95 % Confidence interval, * ORs adjusted for age (continuous), gender, basic education, marital status, BMI, smoking, leisure time physical activity, alcohol dependence, job strain, social support at work, sleep problems, musculoskeletal disorders, mental disorders, and any other diseases.

**Table 3 ijerph-16-02099-t003:** Sum of risk and protective factors in association with sickness absence trajectories. Odds ratios (OR) from multinomial regression analyses with the low sickness absence trajectory as reference.

Summary Exposure	All	Low	Slowly Increasing	High	Full Model *		Full Model *	
					Trajectory Slowly Increasing vs. Low		Trajectory High vs. Low	
		*N*	*N*	*N*	OR ^α^	95% CI ^β^	OR ^α^	95% CI ^β^
Factors that decrease the risk of sickness absence								
Prolonged sitting or keyboard use								
Neither	2207	1130	784	293	1		1	
Either	763	449	241	73	0.79	0.65–0.95	0.66	0.49–0.88
Both	844	504	262	78	0.73	0.61–0.87	0.57	0.43–0.77
Number of factors that increase the risk of sickness absence (nine work factors ^€^)								
0	1258	770	391	97	1		1	
1	992	541	346	105	1.23	1.02–1.48	1.39	1.02–1.90
2−3	769	405	260	104	1.24	1.01–1.52	1.84	1.34–2.53
≥4	795	367	290	138	1.54	1.25–1.89	2.71	1.99–3.69

^α^ Odds ratio, ^β^ 95 % Confidence interval, * ORs adjusted for age (continuous), gender, basic education, marital status, BMI, smoking, leisure time physical activity, alcohol dependence, job strain, social support at work, sleep problems, musculoskeletal disorders, mental disorders, and any other diseases. ^€^ Prolonged standing, repetitive arm movement, arms above shoulder level, bent postures, squatting or kneeling, using a vibrating tool, high hand grip force, frequent handling of loads at least 5 kg, handling of loads at least 20 kg.

**Table 4 ijerph-16-02099-t004:** Combination of risk and protective factors in association with sickness absence trajectories. Odds ratios (OR) from multinomial regression analyses with the low sickness absence trajectory as reference.

Combined Exposure	Low	Slowly Increasing	High	Full Model *		Full Model *	
				Trajectory Slowly Increasing vs. Low		Trajectory High vs. Low	
	*N*	*N*	*N*	OR ^α^	95% CI ^β^	OR ^α^	95% CI ^β^
All (*N* = 3814)							
Prolonged sitting or keyboard use or physically demanding work factors ^€^							
Neither	290	169	48	1		1	
Prolonged sitting or keyboard use only	480	222	49	0.78	0.61−1.01	0.60	0.39−0.94
Physically demanding work factors ^€^ only	840	615	245	1.23	0.98−1.56	1.59	1.11−2.26
Both	473	281	102	0.98	0.77−1.25	1.13	0.76−1.67

^a^ Odds ratio, ^β^ 95 % Confidence interval, * ORs adjusted for age (continuous), gender, basic education, marital status, BMI, smoking, leisure time physical activity, alcohol dependence, job strain, social support at work, sleep problems, musculoskeletal disorders, mental disorders, and any other diseases. ^€^ Prolonged standing, repetitive arm movement, arms above shoulder level, bent postures, squatting or kneeling, using a vibrating tool, high hand grip force, frequent handling of loads at least 5 kg, handling of loads at least 20 kg.

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
