# Peer review of "Work-Related Exposures and Sickness Absence Trajectories: A Nationally Representative Follow-up Study among Finnish Working-Aged People"

_ijerph, 2019, doi:10.3390/ijerph16122099_

Round 1
Reviewer 1 Report
This is an excellent paper, very well-written, easy to read and of great interest. The research question is well formulated, the methods section is detailed enough to warrant reproducibility and to assess the validity of results. The methodology is sound and adequate to address the questions posed with sifficient validity. Results presentation is concise and well-explained. The authors have discussed all major issues regarding methodology and results. The conclusions are drawn from the results with no further speculations. This work deserves publication.
There are however a couple of minor issues which should be addressed before publishing.
In the discussion, the authors address the issue of a potential healthy worker effect (page 9 lines 21-27). The authors should address here in which way this potential bias could have had influenced their results, i.e. do they expect an overstimation or an underestimation of the effects observed? In line 26 the ey state the "risk of sciatica was found for women", I am afraid something is missing here...
The authors adjusted for all relevant factors in their analysis, including muskuloskelettal disease. In my opinion MSD may be a consequence of he exposure and the reason for sickness leave (as the authors themselves state in page 9 line37), thus the reader could think they overadjusted. I think the authors should thus discuss this and reason/explain why is adjusting for MSD in their analysis adequate.
p. 2 l. 44: please provide the direct link to methodology in english. The link provided here redirects to https://thl.fi/fi/etusivu in Finnish. The link provided in reference 22 unfourtunately does not work.
p. 3 l.4: Do you mean all past jobs or only the last one? Reading the whole paper I understood that you asked for the entire exposure biography? Am I right? Please express a little bit clearer
p. 2 l. 48. Maybe "link function zip", which Software?
p. 5 l. 22 I guess "3.1. Subsection" can be deleted...
p. 8 l. 7, the € sign is missing in the table.
p. 8 l. 27, I guess "were" at the end of the sentence can be deleted, or something else is missing?
p. 9 l.10 "to" is missing between "compared" and "manual"
p. 9. l.13 and p. 10 l. 16 "si@ng" probably you mean "sitting"
p. 10 l. 26 "the risk of sciatica was ... found for women" I guess here is soemthing missing ("to be higher" or similar?)
Author Response
*************************************************************** This is an excellent paper, very well-written, easy to read and of great interest. The research question is well formulated, the methods section is detailed enough to warrant reproducibility and to assess the validity of results. The methodology is sound and adequate to address the questions posed with sufficient validity. Results presentation is concise and well-explained. The authors have discussed all major issues regarding methodology and results. The conclusions are drawn from the results with no further speculations. This work deserves publication. There are however a couple of minor issues which should be addressed before publishing. In the discussion, the authors address the issue of a potential healthy worker effect (page 9 lines 21-27). The authors should address here in which way this potential bias could have had influenced their results, i.e. do they expect an overestimation or an underestimation of the effects observed? In line 26 they state the "risk of sciatica was found for women", I am afraid something is missing here... *************************************************************** RESPONSE: Many thanks for the very clear and concrete comments and nice review. The detailed comments and noticing the typos are much appreciated! We have clarified the discussion, regarding the assumed influence of healthy worker effect on our results, and about the results of the earlier study. Please see page 11: “Healthy worker effect could have made our results more conservative.” *************************************************************** The authors adjusted for all relevant factors in their analysis, including musculoskeletal disease. In my opinion MSD may be a consequence of the exposure and the reason for sickness leave (as the authors themselves state in page 9 line37), thus the reader could think they overadjusted. I think the authors should thus discuss this and reason/explain why is adjusting for MSD in their analysis adequate. *************************************************************** RESPONSE: Thank you for pointing this out, it is a valid comment. We have examined all associations without adjustment for musculoskeletal disorders, and the results are very similar. We have modified the discussion to describe the results of the additional analyses that confirm the original results on pages 11: “One may also question, if musculoskeletal disorders should be adjusted for, if sickness absence is largely due to these diagnoses, i.e., should such an adjustment be considered as over-adjustment. We have, however, conducted additional analyses without adjustment for musculoskeletal disorders, and the results were very similar (please see Table S3 and S4 which repeat analyses of the main tables 3 and 4 without adjustment for musculoskeletal disorders). Overall, odds ratios for sitting and computer work attenuated slightly, and those for physical work-exposures strengthened. However, statistical significance did not change for any of the exposures except for bent postures (all odds ratios were significant). Thus, over-adjustment should not be a major issue, or bias the examined associations. As we have several predictors and present different sets of analyses, we preferred to retain full models in the main tables, considering key pertinent risk factors of sickness absence.” We feel this is an important clarification, as readers of the paper could wonder the same question. As we had such a large number of predictors and analyses, showing both crude and adjusted associations in the tables would have made them very busy though. *************************************************************** p. 2 l. 44: please provide the direct link to methodology in English. The link provided here redirects to https://thl.fi/fi/etusivu in Finnish. The link provided in reference 22 unfortunately does not work. *************************************************************** RESPONSE: Thank you. We have corrected the reference and link, it had been changed. The report can be freely downloaded and can be easily found online using the reference given, or from: https://www.julkari.fi/bitstream/handle/10024/78185/2008b26.pdf?sequence=1&isAllowed=y *************************************************************** p. 3 l.4: Do you mean all past jobs or only the last one? Reading the whole paper I understood that you asked for the entire exposure biography? Am I right? Please express a little bit clearer *************************************************************** RESPONSE: We have clarified the measurement of past jobs, as requested. Thus, current job and past five jobs are included in the history, not just the last one. The idea was to focus on specific exposures, thus even if respondents had changed jobs, they might have similar exposures that are important to consider. See page 3: “Participants reported whether they were exposed daily to twelve exposures (no/yes) in their current or past jobs,…” Similarly, the sickness absence follow-up is from any jobs during the follow-up, as the data are from the national registers that gather all sickness absence days lasting more than 9 days, independent of the employer. See page 3: “SA data were obtained from the Social Insurance Institution of Finland that registers absence periods over nine days from all employers.” *************************************************************** p. 2 l. 48. Maybe "link function zip", which Software? *************************************************************** RESPONSE: We have clarified the statistical method section on page 4: “The annual number of SA periods was modelled using zero inflated Poisson distribution (link function Zero Inflated Poisson, ZIP).” ZIP is short for zero inflated Poisson, and the link function is not specific to a software, i.e., several statistical software allow to fit models with ZIP. We used SAS. *************************************************************** p. 5 l. 22 I guess "3.1. Subsection" can be deleted... *************************************************************** RESPONSE: This has been deleted. Apologies for the mistake. It had probably accidently been left from the template that was used. *************************************************************** p. 8 l. 7, the € sign is missing in the table. *************************************************************** RESPONSE: This has been added, thank you for noticing. *************************************************************** p. 8 l. 27, I guess "were" at the end of the sentence can be deleted, or something else is missing? *************************************************************** RESPONSE: Absolutely true. It has been deleted as redundant. *************************************************************** p. 9 l.10 "to" is missing between "compared" and "manual" *************************************************************** RESPONSE: We have added ‘to’ after compared. *************************************************************** p. 9. l.13 and p. 10 l. 16 "si@ng" probably you mean "sitting" *************************************************************** RESPONSE: This has been corrected, thank you for noticing! *************************************************************** p. 10 l. 26 "the risk of sciatica was ... found for women" I guess here is something missing ("to be higher" or similar?) *************************************************************** RESPONSE: We have revised the sentence to be clearer on page 11: “…where participants who had a history of physically heavy work and had exited paid employment, had a higher risk of sciatica [12"].Reviewer 2 Report
The aim of the study is interesting, reporting about a field of public health such as the sickness absence trajectories, not very often considered regarding to physical work-related factors. Each part of the manuscript is good, in particular about materials and methods. Therefore, I have no remarks to make.
Author Response
***************************************************************** The aim of the study is interesting, reporting about a field of public health such as the sickness absence trajectories, not very often considered regarding to physical work-related factors. Each part of the manuscript is good, in particular about materials and methods. Therefore, I have no remarks to make. ***************************************************************** RESPONSE: Thank you very much for this most positive and nice review! We very much appreciate your kind words.Reviewer 3 Report
Abstract- This should be more specific. It is poorly presented.
Introduction:
Objective/Gap: the need for this research is not clearly stated. The main objectives of the study are not clear. The main research questions are not well developed.
Theoretical development:
The paper did not discuss any theory to support arrangements. the theoretical assumptions of the research questions development are confusing and are not well justified.
Methodology: This paper does not contain any specific methodology.
Discussion – this should include an account of the results obtained in relation to the aims and hypotheses of the study, the possible strengths and weaknesses of the study and a comparison to similar studies in this field.
This manuscript should follow the guideless for authors of the International Journal of Environmental Research and Public Health (see https://www.mdpi.com/journal/ijerph/instructions) and requires editing of the English language and style. Additionally, the use of acronyms is not appropriate throughout the paper.
Although I think the topic raised in this paper is interesting, the paper should be improved substantively before considering a publication. First, as an empirical study, it fails to follow the standard process. Second, you'd better discuss the theoretical gap in the introduction section, and make your study based on solid theoretical foundations. Third, provide sufficient information on the measurement of the key variables, Four, discuss the potential contributions made to the literature and the implications for management before the end. This paper does not significantly contribute to the existing literature and it does not make a sufficiently novel contribution.
Author Response
****************************************************************** Abstract- This should be more specific. It is poorly presented. ****************************************************************** RESPONSE: We have carefully re-read the abstract and checked that it includes all the relevant details. However, some specific details are missing, as this study had a large number of predictors and many analytical steps, whereas abstract word count is very limited (max 200 words). This will be an open access paper so any reader will be able to read further details from the paper. ****************************************************************** Introduction: Objective/Gap: the need for this research is not clearly stated. The main objectives of the study are not clear. The main research questions are not well developed. ****************************************************************** RESPONSE: The aims of the study were found clear by the two other reviewers, and thus we preferred to retain them. We have, however, carefully checked the formulations throughout and aimed to clarify the manuscript throughout where needed and feasible, including the objectives and gaps in current evidence. Examples of the main gaps mentioned in the Introduction: “However, the evidence is still inconclusive and could depend on the used exposures, age groups, and methods.” “Indeed, another important gap in evidence is that the different work-related factors have rarely been studied in combinations in relation to the risk of SA.” “Previous studies have often either measured work exposures only at a cross-sectional time point, or with a long gap between repeated measurements, thus without possibilities to confirm the significance of long-term exposure to SA [4].” “Finally, person-oriented methods have only rarely been applied when examining the associations between physical work-related factors and SA [2,3].” The objectives/aims are formatted as: “…, we first identified SA trajectories over a 7-year follow-up using nationally representative data. Second, we examined whether 12 work-related factors, individually and cumulatively, increase or decrease the risk of SA, in terms of group memberships in the identified SA trajectories..” Theoretical development: The paper did not discuss any theory to support arrangements. the theoretical assumptions of the research questions development are confusing and are not well justified. RESPONSE: It is true that some papers present more theoretical background, particularly in the field of sociology, for example. However, in many epidemiological papers introductions are often quite brief, and the aim is to briefly summarize what is known and what the study aims to add. In observational studies, testing hypotheses or presenting theories is seldom relevant. We thus preferred to retain the introduction relatively short and as a summary of previous work in this area, as well as point out the methodological advancement needed. ****************************************************************** Methodology: This paper does not contain any specific methodology. ****************************************************************** RESPONSE: We have amended the description of the methods in this revised version of our manuscript. We present the methods used in the introduction, methods section as well as discuss their strengths and limitations in the discussion. The methods are not novel (which would be rare in any paper except for methodological ones) but they have quite seldom been used in this area, and therefore help produce more information as compared to previous studies. ****************************************************************** Discussion – this should include an account of the results obtained in relation to the aims and hypotheses of the study, the possible strengths and weaknesses of the study and a comparison to similar studies in this field. ****************************************************************** RESPONSE: Our discussion is built by first summarizing the main findings as a reflection of the aims, followed by comparison to previous studies, and then considering the methodological strengths and limitations. We prefer to avoid too strong conclusions not supported by the data. ****************************************************************** This manuscript should follow the guideless for authors of the International Journal of Environmental Research and Public Health (see https://www.mdpi.com/journal/ijerph/instructions) and requires editing of the English language and style. ****************************************************************** RESPONSE: We have prepared the manuscript following the journal guidelines and carefully applied the template provided at the journal website. Using the template helped insure that the font, layout, and all the sections including presentation of the tables and figures follow the journal guidelines. ****************************************************************** Additionally, the use of acronyms is not appropriate throughout the paper. ****************************************************************** RESPONSE: We have checked the used acronyms throughout are used in a consistent way and are appropriate. ****************************************************************** Although I think the topic raised in this paper is interesting, the paper should be improved substantively before considering a publication. First, as an empirical study, it fails to follow the standard process. Second, you'd better discuss the theoretical gap in the introduction section, and make your study based on solid theoretical foundations. Third, provide sufficient information on the measurement of the key variables, Four, discuss the potential contributions made to the literature and the implications for management before the end. This paper does not significantly contribute to the existing literature and it does not make a sufficiently novel contribution. ****************************************************************** RESPONSE: Thank you for your comments. We have aimed to response to all these comments in our point by point response above, and we have revised the manuscript where feasible. Measurement of covariates is reported in more detail. We have also checked the English language throughout.